# Effect of Transcutaneous Spinal Direct Current Stimulation in Patients with Painful Polyneuropathy and Influence of Possible Predictors of Efficacy including BDNF Polymorphism: A Randomized, Sham-Controlled Crossover Study

**DOI:** 10.3390/brainsci13020229

**Published:** 2023-01-30

**Authors:** Hedayat Rahin, Walker Scot Jackson, Magnus Thordstein

**Affiliations:** 1Department of Clinical Neurophysiology, Region Östergötland University Hospital, 58185 Linköping, Sweden; 2Department of Clinical and Experimental Medicine, Linköping University, 58185 Linköping, Sweden; 3Department of Biomedical and Clinical Sciences, Linköping University, 58185 Linköping, Sweden

**Keywords:** neuromodulation, trans-spinal direct current stimulation, tsDCS, neuropathic pain, BDNF

## Abstract

*Introduction*: The neuromodulating effects of transcutaneous-spinal Direct Current Stimulation (tsDCS) have been reported to block pain signaling. For patients with chronic pain, tsDCS could be a potential treatment option. To approach this, we studied the effect of anodal tsDCS on patients with neuropathic pain approaching an optimal paradigm including the investigation of different outcome predictors. *Methods*: In this randomized, double-blinded, sham-controlled crossover study we recruited twenty patients with neurophysiologically evaluated neuropathic pain due to polyneuropathy (PNP). Variables (VAS; pain and sleep quality) were reported daily, one week prior to, and one week after the stimulation/sham period. Anodal tsDCS (2.5 mA, 20 min) was given once daily for three days during one week. BDNF-polymorphism, pharmacological treatment, and body mass index (BMI) of all the patients were investigated. *Results*: Comparing the effects of sham and real stimulation at the group level, there was a tendency towards reduced pain, but no significant effects were found. However, for sleep quality a significant improvement was seen. At the individual level, 30 and 35% of the subjects had a clinically significant improvement of pain level and sleep quality, respectively, the first day after the stimulation. Both effects were reduced over the coming week and these changes were negatively correlated. The BDNF polymorphism Val66Met was carried by 35% of the patients and this group was found to have a lower general level of pain but there was no significant difference in the tsDCS response effect. Neither pharmacologic treatment or BMI influenced the treatment effect. *Conclusions*: Short-term and sparse anodal thoracic tsDCS reduces pain and improves sleep with large inter-individual differences. Roughly 30% will benefit in a clinically meaningful way. The BDNF genotype seems to influence the level of pain that PNP produces. Individualized and intensified tsDCS may be a treatment option for neuropathic pain due to PNP.

## 1. Introduction

Neuropathic pain is widely considered to be one of the most challenging pain conditions to treat [1]. Although this number is believed to be underestimated, recent epidemiological studies indicate that neuropathic pain has a prevalence of 3–10% in the general population [1,2]. Various causes can induce the initial damage to nervous tissue, either in the peripheral or in the central nervous system, that may lead to neuropathic pain [2]. Yet, the clinical presentation in patients is similar, with the pain typically described as a burning and sharp pricking or stabbing sensation, often combined with other sensory and/or motor nerve deficits [2,3]. The pathophysiological background to neuropathic pain is complex and still not fully known. Peripheral and central processes, deafferentation, as well as neurogenic inflammation, is believed to lead to a central neural sensitization with enhanced excitatory and/or reduced inhibitory transmission in pain-controlling systems [2,3].

Polyneuropathy (PNP) is a progressive condition where multiple peripheral sensory and/or motor nerves are affected, resulting in varying degrees of muscle weakness, sensory impairments, and often neuropathic pain [4]. Patients with PNP account for a vast portion of people suffering from chronic neuropathic pain. With the pain often located symmetrically to the lower extremities/feet, PNP is a common complication in diseases and disorders such as diabetes mellitus, alcoholism, and some hereditary disorders. However, in most cases the cause is unknown [2,4].

Generally, pharmacological treatment for neuropathic pain at best only offers partial pain relief and often has unmanageable side effects. Thus, today’s treatment outcomes are often inadequate and unsatisfactory [1,2]. 

Neuromodulation is a rapidly growing field in medicine. Conceptually, it includes different stimulation techniques to alter or counteract various imbalances of the central, peripheral, or autonomic nervous systems [5]. With the aim to treat neuropathic pain due to PNP, we have studied one of the more novel, non-invasive paradigms; transcutaneous spinal direct current stimulation (tsDCS). The technique involves a low-intensity spinal stimulation and has been demonstrated to induce polarity-dependent effects on different pathways of the spinal cord, e.g., [6,7,8]. One promising effect is that of anodal tsDCS on pain signaling; in a double-blind, sham-controlled study on healthy subjects we have been able to reproduce and expand on prior studies [6,7] finding parallel inhibitory effects of anodal tsDCS on subjective pain experience and pain signaling [9]. 

A major obstacle for neuromodulation techniques is the large interindividual variation regarding stimulation effects [6,7,8,9]. This necessitates that evaluation of treatments is made not only at the group, but also at the individual level. One possible source for this variability is a common single nucleotide polymorphism (SNP) in the gene encoding brain-derived neurotrophic factor (BDNF). This SNP results in the replacement of Val 66 by Met and the production of a less effective BDNF protein. BDNF is believed to influence several important neuronal processes and the polymorphism has been associated with complex changes in BDNF function [10]. One of these functions is synaptic plasticity and recent findings indicate that the BDNF genotype of an individual can prognosticate the response to tsDCS [11]. Indeed, we too have found that healthy participants with the BDNF Val66Met genotype respond less well to the pain-reducing effects of anodal tsDCS [12]. Most of the current evidence [13] regarding pain perception and BDNF point towards a pro-nociceptive mechanism, although newer findings question this claim [10]. Other factors, such as pharmacological interactions [14] and proportion of subcutaneous fat tissue [15] could potentially be part of predicting the effects of DC stimulation and contribute to the interindividual variability. 

In the present study, we aimed to evaluate the effect of anodal tsDCS on PNP patients with neuropathic pain located in the feet/lower extremities. The protocol that was tested was one that would be reasonable in clinical practice (three visits per week). We genotyped the BDNF and evaluated the effect of pharmacological treatment and body size. In doing so, we sought a possible way to optimize, possibly individualize, the treatment protocol for tsDCS in this group of patients.

## 2. Materials and Methods

A total of 21 patients with neuropathic pain affecting lower extremities were recruited for this study. Information regarding weight, height and pharmacological treatments were gathered at the start of the study. All patients were recruited from the neurophysiological clinic in Linköping, Sweden, and had previously performed quantitative sensory testing (QST) between 2019 and 2022. The study was registered in research-web.org.

### 2.1. Study Design

This was a randomized, double-blind, sham-controlled crossover study (Figure 1). All subjects participated over a seven-week period and were randomized to start either with sham or real tsDCS. Simple randomization was used through a computer-generated model on MATLAB (version 9.2, R2017a, The MathWorks, Inc., Natick, MA, USA). For the stimulation, two DC stimulators (Sooma Oy, Helsinki, Finland) were used, one for verum and one for sham stimulation. The stimulators were prepared by an engineer and labelled “A” and “B”. Thus, both investigators and patients were blinded. The code was broken after the treatment of the last patient. Patients were assigned to “A” or “B” on the first treatment session.

Evaluations were made using visual analogue scales (VAS, cf. below) one week prior to, and one week after the stimulation/sham period. The subjects were used as their own controls, hence before gathering the baseline values for the second stimulation period, there was a washout period of one week. 

### 2.2. Eligibility Criteria for Participants

Patients between ages 18 and 85 with clinically-proven, therapy-resistant, painful polyneuropathy from the feet were eligible for the study. All patients with deviating QST test results from the feet, compatible with small fiber neuropathy [16] were recruited. Prior to the study start, all the subjects were further screened for neuropathic pain though the painDETECT questionnaire (PD-Q) [17], and those with a score of at least 13 (of maximally 38) were included in the study. In the PD-Q, a result less than 13 is considered negative, indicating that a neuropathic component to the pain is unlikely. The exclusion criteria were severe illness (e.g., heart failure, kidney failure), implanted electrical devices (e.g., spinal cord stimulator, pacemaker), pregnancy, and if any new treatments for neuropathic pain had started within two months prior to study start or if such treatment was changed during the testing period.

### 2.3. Ethical Statement

The protocol was approved by the Ethical Review Board and all participants signed forms accepting participation prior to the start of the study.

### 2.4. Anodal Stimulation; tsDCS

For anodal stimulation, a constant electrical current of 2.5 mA (total charge of 63.9 mC/cm^2^) was delivered for 20 min, whereas the sham stimulator did so for only 20 s before returning to zero. The current was delivered through two electrodes (rubber, 7 × 5 cm, longer side centralized over and along the spine, placed in sponges soaked in physiological saline). Automatic control of resistance would have hindered stimulation at a resistance above 15 mΩ. The electrodes were placed and fixed with adhesive tape with the subject in a prone position. From the edges of the electrodes, the anode was positioned 7.5 cm caudally and cathode 7.5 cm cranially to the spinous process of the 10th thoracic vertebra, i.e., 15 cm between the edges of the electrodes. Stimulations were given on three days (Monday, Wednesday, and Friday) for one week. During the stimulation, the subjects were questioned every 5 min if they had a perception from the electrodes.

### 2.5. BDNF Genotyping

Buccal cells from the participants were captured through a rinse with phosphate buffered saline and then concentrated by 3000× *g* centrifugation. DNA from the cells was extracted following proteinase-K digestion and ethanol precipitation. The samples were genotyped by PCR amplification using an established protocol [18] and analyzed by agarose gel electrophoresis. The result that was produced was a typing of the patients to one of three genotypes (Val/Val, Val/Met, and Met/Met).

### 2.6. Outcome Variables

The subjects scored symptoms with VAS twice a day (morning/evening) for a week before and after each stimulation week. The parameters that were used were level of general, pin pricking, and burning pain (scale for all 0–10; 0: none, 10: extreme) and sleep quality (0–10; 0 none, 10: extremely good). The mean score for morning/evening was used to represent each day. Calculations for a sham-corrected effect was made for the results ([(baseline 1-sham)-(baseline 2-real tsDCS)]). Body mass index (BMI) was calculated according to the formula mass (kg)/height2 (m). Pharmacological treatments with potential mechanism on neuropathic pain were noted on each patient. During the analyses, patients were categorized according to their BDNF genotype, BMI, and if they had any pharmacological treatment for neuropathic pain and then compared to their counter group with regards to the stimulation effect. 

### 2.7. Statistical Analysis

Baseline values for all the recorded parameters are presented as means and standard deviations. Shapiro–Wilk’s test was used to verify a normal distribution for all the results. An independent *t*-test was used to test for differences between the baseline levels before the first and second stimulation periods. A pre-test for carryover effect were made according to the method described by Wellek et al. [19]. The results after real and sham stimulation are presented in relative terms as percentages of baseline levels. A pain reduction and sleep quality improvement of 30% were considered clinically significant [20]. A *p*-value less than 0.05 was considered significant.

The differences between the effects of real vs. sham tsDCS for each subject were calculated. These differences are presented at both the group and individual level. Paired *t*-tests were used to determine significant changes over time for the groups comprising all individuals, while we employed independent *t*-test to compare the subgroups (BDNF genotypes, pharmacological treatment, and BMI). Finally, Pearson’s correlation calculation was used to test for correlation between general pain reduction and sleep quality. The analyses were made using IBM SPSS Statistics (Version 27.0, IBM Corp, Armonk, New York, NY, USA) and GraphPad Prism (version 9.0, GraphPad Software, San Diego, CA, USA). 

## 3. Results

The results are presented for 20 of the patients. One participant was excluded because of incomplete results. Their ages were between 36 and 82 (mean, SD 62.9 ± 12.7, 9 female). Demographics and BDNF genotypes for all the participants are presented in Table 1. Data from days 1, 3, 5, and 7 after the stimulation week are presented (data from the intervening days did not change the overall image). Saliva samples for BDNF-analysis were taken from all patients but one, who was unable to provide this. A total of 40% had no pharmacological treatment for pain. Patients were asked post-sham/real stimulation to guess the type of stimulation, 35% guessed correctly. 

During the real tsDCS stimulations, 90% of the participants felt a weak tingling or burning sensation under the anode and/or cathode, mainly at the start of the stimulations. For sham stimulation, 75% experienced the same. A very slight redness was noticed under the cathode and/or anode after all stimulations for both real and sham tsDCS. All participants tolerated the treatment well and no other adverse effects were reported. Stimulation order and baseline values for all parameters are presented in Table 2. No significant difference was found between baseline values for any of the parameters. Pre-tests showed no significant carryover effect (*p* = 0.56), indicating that a sufficient washout period was used. 

### 3.1. Group Values Regarding tsDCS Effects

Here, we compared the baseline corrected results of all patients, grouped to the response after sham and real stimulation. Using a paired *t*-test, no significant difference was found between day 1 and day 7 for general pain (*p* = 0.12, Figure 2). For sleep quality (Figure 3), the results were more apparent with a significantly reduced sleep quality over the week of observation (the difference between the means of day 1 and day 7 was significant: 26 percentage units, *p* < 0.05). Thus, for both parameters, trends of a vanishing effect over time were seen. There was a moderately strong negative correlation (−0.53, *p* < 0.05), between these two parameters.

### 3.2. Effects of tsDCS on General Pain Regarding BDNF Polymorphism, Pharmacological Treatment, and BMI

A total of 35% (7/20) of the patients had Val66Met polymorphism (Table 1) (none had the Met/Met polymorphism). Polymorphism did not influence the effect of stimulation as no significant difference was found (Figure 4). However, baseline values of pain for both such weeks for the patients with Val/Met were significantly lower compared to those of the Val/Val group. Mean baseline values (VAS) for the Val/Val group was 7,27 (week 1) and 7.35 (week 5), compared to 4.39 (baseline 1; *p* < 0.001) and 4.79 (baseline 2; *p* < 0.001) for the Val/Met group.

Half of the patients had pharmacological treatment for neuropathic pain (Table 1). No significant difference regarding the effect of stimulation could be found between patients with or without pharmacological treatment (Figure 4). The BMI results for all patients are given in Table 1. A total of 40% (8/20) had a BMI < 25, 35% (7/20) BMI 25–30, and 25% (5/20) BMI > 30. There was a non-significant tendency for a better effect for the patients with lower BMI, especially over time (Figure 4).

### 3.3. Individual Values Regarding tsDCS Effects

The individual baseline corrected differences between the effects of sham and real stimulation [(baseline 1-sham) − (baseline 2-real tsDCS)] for the four evaluated parameters are given in Figure 5 (calculated in this manner, a larger effect of real stimulation produces a negative value). For all the pain parameters, we observed a general tendency towards a reduction of the pain levels with real tsDCS compared to sham. At day 1, 30% (6/20) of the patients had a sham corrected, clinically significant general pain reduction (≥30% below baseline). Over time, by day 7, a reduction of this number was seen to 20% (4/20). Similar results were seen for the other pain parameters. Notably, only one of the patients with the Val/Met polymorphism reached a clinically significant general pain reduction (days 1, 3, and 7). 

The results regarding sleep quality show that at day 1 35% (7/20) of the patients experienced a sham corrected, clinically significant improvement (≥30% above baseline) with real tsDCS. In parallel to pain sensations, the effect declined over the week after stimulation to 15% (3/20) at day 7. Remarkably, a few patients reported a greater effect from sham stimulation compared to real tsDCS.

## 4. Discussion

Non-invasive neuromodulation techniques including tsDCS, have been shown to affect different aspects of central nervous system functioning, both at the cerebral [5] and at the spinal level [21]. The present study aimed to investigate a strategy, possibly for clinical use, applying tsDCS as a treatment for patients with painful polyneuropathy of lower extremities. We investigated a group of patients with therapy refractive pain, using currently available pharmacologic and physiotherapeutic treatments. An important aspect for transfer to clinical practice was to include possible individual factors that are predictive of a positive outcome. We used a careful methodology in a double blind, sham-controlled crossover study design with consideration to the fluctuating levels of neuropathic pain [22]. With the low intensity paradigm that was used in this study (three simulations, one every other day during five days; 2.5 mA 20 min) we found trends to, but no significant effects on pain levels at the group level. However, a clinically relevant reduction of pain was seen in 30% of the patients. The largest effect for all parameters was recorded at the first day after the treatment week, although 20% of the patients still had a clinically significant pain reduction by day 7. In conjunction with the reduced pain, a significant improvement in sleep quality was seen for 35% (first day) of the patients. This fits with reports of interference of pain on sleep being one of the most disabling aspects for these patients [4]. Our findings highlight the importance of evaluating the effects of neuromodulation not only at the group but also at the individual level.

The exact mechanism of tsDCS remains to be clarified. However, it is apparent that the effects on the spinal cord are polarity dependent, and anodal stimulation has been shown to inhibit nociceptive responses [6,7,8,9]. Effects have been shown to occur on both spinal and supraspinal levels [21,23], affecting nerve fiber excitability and synaptic transmission [23]. For patients with neuropathic pain, partially induced by a neurogenic inflammation [3], a likely effect of tsDCS could be an anti-inflammatory response with pain reducing effects [13,24].

The location for stimulation was chosen based on modelling studies demonstrating that the electric field that is created by tsDCS is dependent on the location of the electrodes with field effects on several segments and pathways through the depth and length of the spinal cord [15]. At the lumbar level, the highest field strength is produced halfway between electrodes when these are placed along the spinal cord [15], as in the present study. Placing the electrodes at equidistance, in our case 7.5 cm, above (cathode) and below (anode) the spinous process of the vertebra Th10, is believed to aim the focus point at the level of the dorsal roots from the afferent fibers from the feet and legs to the spinal levels L5-S1 [15]. Another question that needs to be considered is the dosing of the treatment. Early studies demonstrated short-lasting modulating effects from a single stimulation [6,7,9], and only recently, repeated sessions during a week were shown to produce a more long-lasting effect [8]. Our results demonstrate that for 30–35% of these patients, one stimulation with tsDCS every other day during five days, is well tolerated and produces clinically relevant effects for up to one week. We propose that a more intense stimulation strategy is likely to improve the effects. Although the sham stimulation that was used in this study has been proven to not exert any clinical or physiological effects [6,7,8], several of the patients experienced a slight redness on the skin under the electrodes after both sham and real stimulation. This could possibly be a dermal reaction to the saline-soaked sponges. 

This study replicated the findings from numerous previous investigations [8,9], i.e., we found large interindividual variability regarding the effect of neuromodulation. In our opinion, there are two aspects that need to be considered to overcome this obstacle to the introduction of neuromodulation into clinical practice. First, grouped results must be recognized to be an insufficient way to interpret the effects of stimulation. Thus, aiming for a clinical setup, analysis of individual data needs to be included. Second, there is a need to find indicators at the individual level that can inform on the probability of therapeutic success. In the present study, we included three such possible indicators; BDNF genotype, phenotype, and ongoing pharmacological treatment. 

At the group level, there were no significant differences between the two studied BDNF genotypes (Val/Val, Val/Met). At the individual level, however (Figure 5), we found a stronger stimulation effect for the patients with the most common type; Val/Val, as previously reported [11,12]. Interestingly, the baseline pain level for patients with the Val/Met SNP was significantly lower than for the patients with Val/Val. To the best of our knowledge, this effect of BDNF polymorphism has not been previously reported. One possible explanation for this might be the novel findings of the possibility of both pro- and antinociceptive mechanisms of the BDNF protein [10]. Taken together, individuals with the Val/Met genotype might be less prone to develop severe chronic pain, but if they do, treatment with neuromodulation may be less effective. 

Individual phenotype might also be a prognostic factor. Fat tissue has lower electrical conductivity compared to muscle tissue [15,25]. Hence, the proportions of different tissues might influence the strength of the electric fields that are induced in the neural structures. The body mass index (BMI) correlates to the amount of subcutaneous fat in general [26]. Accordingly, our results demonstrated a tendency for a greater response of tsDCS for patients with lower BMI compared to those with higher. Thus, BMI could be of importance when choosing treatment intensity. 

Another factor to consider is pharmacological treatment. Previous studies have shown that pharmacological interactions affect the stimulation effects of tDCS [14,27]. Typical pharmaceuticals for neuropathic pain, such as amitriptyline and gabapentin, act on the synaptic transmission [28] and could possibly affect the spinal and supraspinal effects of tsDCS. We could not find any significant effects in this study. This might be due to the fact that only half of the patients had ongoing pharmacological pain treatment. Most of them had previously tried such medications, but a lack of effect and/or side effects had prompted their discontinuation. This underscores the need for new treatments to be developed for this patient group. 

This study does have some limitations. Although individuals with clinically significant improvement were found, we recognize that the small sample size might be a limitation to the conclusions that can come from this study. No sample size calculation was made as this was a first study attempting to evaluate the effect of a structure that might be applicable to clinical routine care. We recruited patients with a broad etiology with regards to their neuropathic pain, which could potentially be a confounding factor to the effect and mechanisms related to tsDCS. Further, different pharmacological treatments for pain were recognized, and the different mechanisms of these medicines could potentially affect the tsDCS effect. 

## 5. Conclusions

Applying a simple arrangement with great potential for home treatment, we show that in patients with polyneuropathy, anodal tsDCS has positive effects on sleep quality and a promising pain reducing effect. The inter-individual variation in response to the stimulation seems to be multifactorial. More studies are needed to optimize the tsDCS protocol for pain treatment and to identify predictor factors. However, after a short trial period, as in this study, responders can already be identified. Therefore, and given its ease of use and lack of serious side effects, trials with more intense protocols employed over longer time in the patient’s home, seem warranted. 

## Figures and Tables

**Figure 1 brainsci-13-00229-f001:**
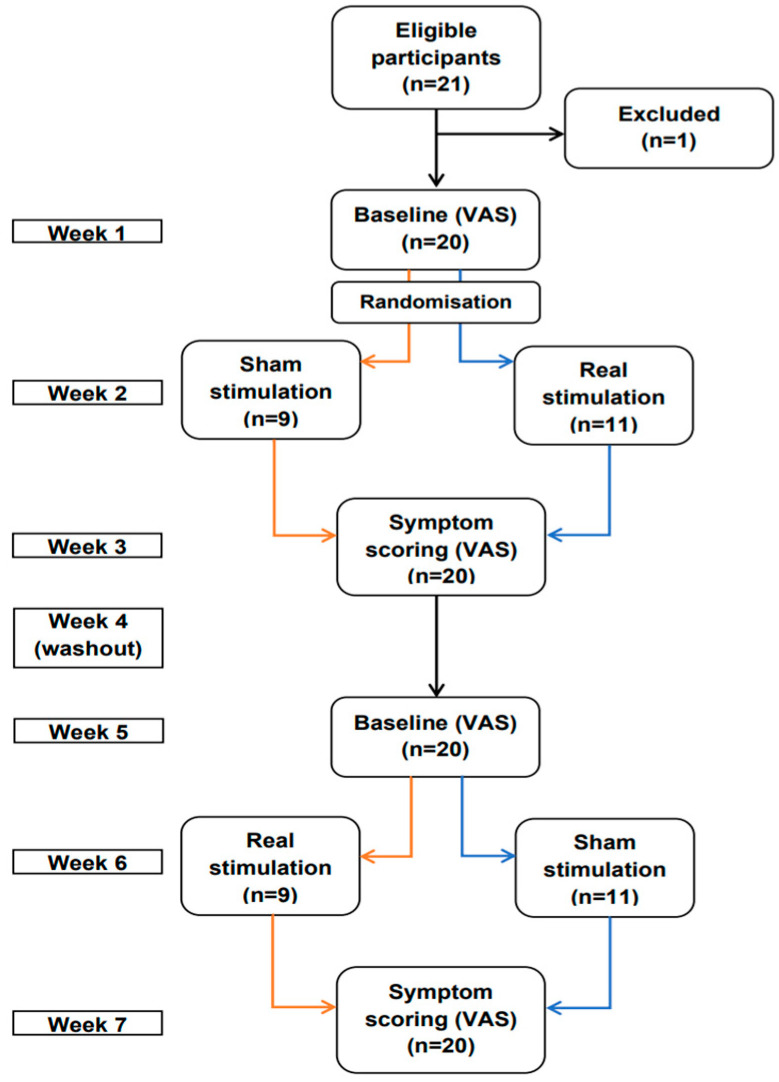
Flowchart of study design.

**Figure 2 brainsci-13-00229-f002:**
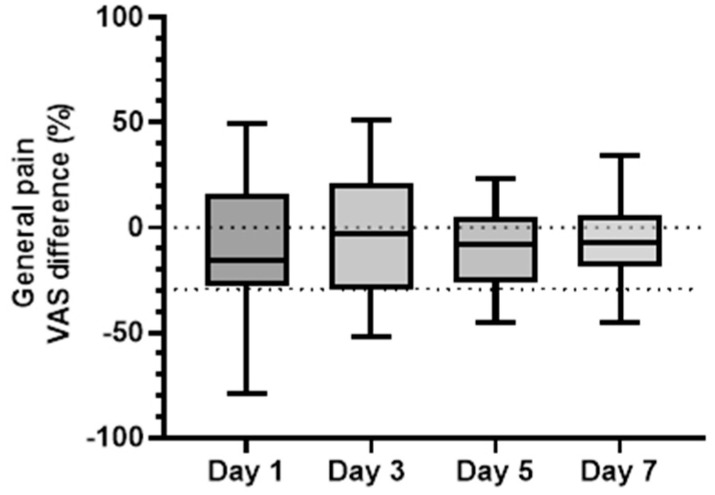
Boxplot presentations (minimum, first quartile, median, third quartile, and maximum) of the difference in general pain levels (VAS, %), group values, between real and sham stimulation during four of the days of the week after stimulation. Negative values indicate a larger effect of real than of sham stimulation. 0% indicates no difference from baseline. A clinically significant level of change (30%) is represented by the grey dotted line.

**Figure 3 brainsci-13-00229-f003:**
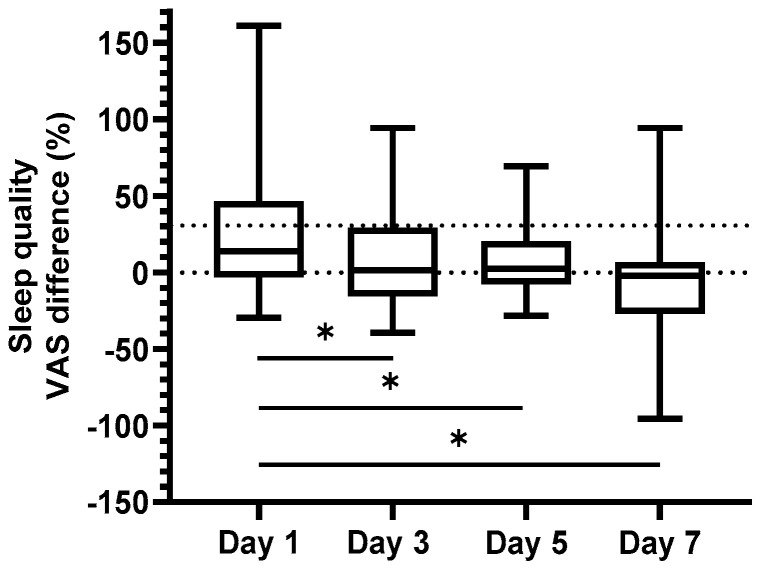
Difference in sleep quality (VAS, %) between real and sham stimulation presented for all participants during four of the days of the week after stimulation. Boxplot characteristics as in Figure 2. Positive values indicate a larger effect of real than of sham stimulation. 0% indicates no difference from baseline. A clinically significant level of change (30%) is represented by the grey dotted line. * = *p* < 0.05.

**Figure 4 brainsci-13-00229-f004:**
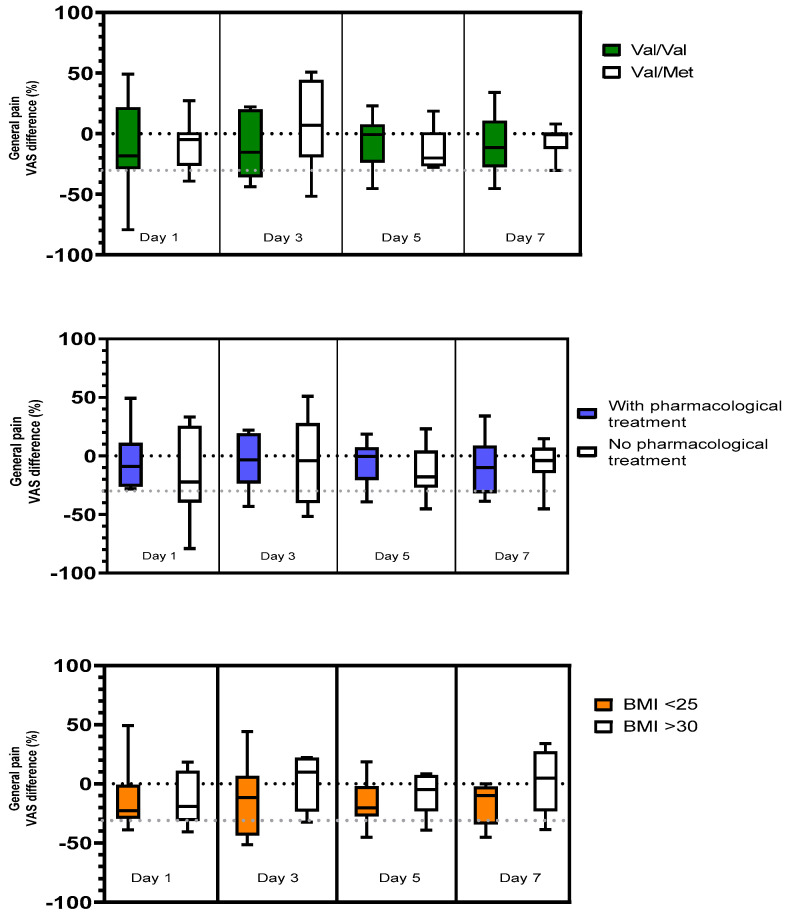
Difference in general pain levels (VAS, %) between real and sham stimulation presented for all participants during four of the days of the week after stimulation. Boxplot characteristics as in Figure 2. Values for patient groups differing regarding BDNF genotypes (top panel), concurrent pharmacological treatment (middle panel), and participants with lower (<25) to higher (>30) BMI (lower panel). Negative values indicate a larger effect of real than of sham stimulation. 0% indicates no difference from baseline. A clinically significant level of change (30%) is represented by the grey dotted lines.

**Figure 5 brainsci-13-00229-f005:**
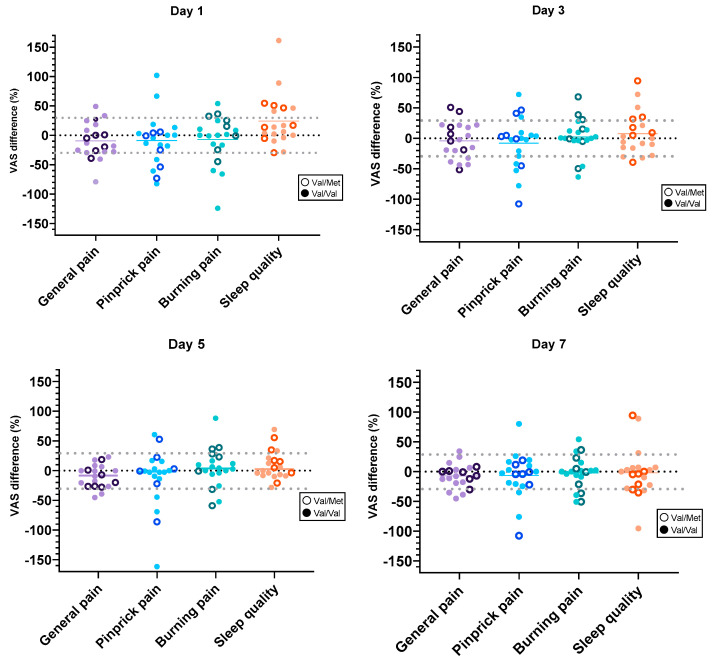
Individual differences for the four evaluated parameters (VAS, %) between real and sham stimulation. A negative value indicates a lower level of reported pain after real compared to sham stimulation. For sleep quality the opposite is true, i.e., a positive value indicates a larger treatment effect after real than after sham stimulation. Clinically significant levels of change (30%) are represented by the grey dotted lines. 0% indicates no difference from baseline. BDNF genotypes marked individually: Val; Valine, Met; Methionine.

**Table 1 brainsci-13-00229-t001:** Demographics and results from pain scoring and BDNF genotyping. BMI, body mass index. PD-Q, PainDETECT Questionnaire. DIPN, drug-induced peripheral neuropathy. DPN, diabetic peripheral neuropathy. LSR, lumbosacral radiculopathy. SLE, systemic lupus erythematosus. Val, Valine. Met, Methionine.

No.	Age	Gender	BMI (kg/m^2^)	Aetiology of Neuropathy	Pharmacological Treatment	PD-Q Score	BDNF Genotype
1	73	Male	24.3	Unknown	Pregabalin	28	Val/Met
2	78	Male	30.4	Unknown	Gabapentin	17	Val/Val
3	36	Female	29.0	Unknown	None	33	-
4	73	Female	23.4	DIPN	None	23	Val/Met
5	52	Male	33.1	DPN	Pregabalin + Tramadol	28	Val/Val
6	71	Female	34.8	Sjögren syndrome	None	18	Val/Val
7	68	Male	25.7	Unknown	None	15	Val/Val
8	78	Female	23.9	Unknown	None	25	Val/Met
9	57	Male	27.1	LSR	None	26	Val/Val
10	68	Female	23.5	Unknown	None	24	Val/Val
11	45	Female	23.5	LSR	Gabapentin	26	Val/Val
12	74	Female	21.1	Unknown	None	22	Val/Met
13	54	Male	30.8	DIPN	Pregabalin + Duloxetine	28	Val/Val
14	60	Male	32.0	Unknown	Amitriptyline	22	Val/Val
15	60	Male	24.6	DIPN	Amitriptyline	22	Val/Met
16	52	Male	34	Unknown	Amitriptyline + Gabapentin	15	Val/Val
17	48	Male	26.0	LSR	None	24	Val/Met
18	82	Female	26.7	SLE	Amitriptyline	18	Val/Met
19	51	Female	28.0	Unknown	None	30	Val/Val
20	78	Male	24.4	Unknown	Gabapentin	28	Val/Val

**Table 2 brainsci-13-00229-t002:** Stimulation order and baseline VAS values with mean, standard deviation, and *p*-values for all parameters.

	Stimulation Order (Start)	General Pain (VAS)	Pinprick Pain (VAS)	Burning Pain (VAS)	Sleep Quality (VAS)
Baseline 1	9/20 (real tsDCS)	6.3 ± 2.1	5.3 ± 2.9	5.3 ± 3.0	4.8 ± 2.5
Baseline 2	11/20 (sham)	6.5 ± 1.7	5.3 ± 2.7	5.6 ± 2.9	4.7 ± 2.5
*p*-value		0.76	0.99	0.83	0.94

## Data Availability

The data that support the findings of this study are available from the corresponding author H.R., upon reasonable request.

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
