# Peer review of "Effect of Transcutaneous Spinal Direct Current Stimulation in Patients with Painful Polyneuropathy and Influence of Possible Predictors of Efficacy including BDNF Polymorphism: A Randomized, Sham-Controlled Crossover Study"

_brainsci, 2023, doi:10.3390/brainsci13020229_

Round 1

Reviewer 1 Report

Review of the manuscript entitled: “Effects of BDNF polymorphism and transcutaneous spinal direct current stimulation in patients with painful polyneuropathy. A randomized, sham-controlled crossover study.”

Thank you for giving me the opportunity to review this fine study examining the effects of BDNF polymorphism and transcutaneous spinal direct current stimulation in patients with painful polyneuropathy. The authors found that tsDCS lead to clinically meaningful amelioration of pain in about 30% pf the patients. The individual BDNF variant seems to influence the level of pain secondary to PNP.

The study is carefully conducted, covering an interesting and important subject. The introduction is straightforward and leads to the hypothesis. The methods are adequate. The results are presented in a comprehensive way. The discussion covers all relevant aspects.

I have some points of criticism/ proposals for changes:

General:

How can it be explained that tsDCS produces clinically relevant effects for up to one week while spinal cord stimulation (SCS) (at least conventional “tonic SCS) which exerts a much more direct stimulation leads to much shorter interval of pain relief after cessation of stimulation and overall pain relief by SCS is higher than in tsDCS (i.e., Pluijms et al 2012)?

Has a sample size been conducted? I guess that the sample size of 20 is sufficient for detecting clinically important differences in pain levels but maybe not for smaller differences in the secondary outcome parameters.

I miss a “limitations section” at the end of the discussion. Here, the sample size should be mentioned. Was, for instance, the stronger stimulation effect for the patients with Val/Val statistically significant?

In the results section and in the discussion I had some difficulties to grasp the meaning of “individual level” and “group level”.

Specific:

Abstract:

L 19/20

“The less common variant of BDNF (Val66Met) was carried 19 by 35 % of the patients and this group had a lower general level of pain.”

Did the Val66Met patients report less pain only at baseline or also in the course of the study?

L20/21

“At the group level, the effects on pain showed tendencies to-, but no significant effects were found when excluding the effect of sham stimulation.”

What is intended with “group level”?

What is intended  with “tendency to-“?  Tendency towards significance?

L22/23

 “At the individual level, 30- and 35 % had a clinically significant (> 30 % change) improvement of pain level and sleep quality respectively the first day after the stimulation.”

What is intended individual level? Please rephrase

Introduction:

L36

“believed to underestimate its occurrence,” please consider rephrasing

Material and Methods:

L93

“Body mass index (BMI) was calculated according to the formula mass (kg)/ height2 (m).”

Maybe this can be deleted as the calculation of the BMI is generally known. Consider giving the unit (kg/m2) within the figure and/or the caption.

L128ff

“From the furthest edge of the electrodes, the anode was positioned 7,5 cm caudally and cathode 7,5 cm cranially”

This does not become quite clear to me. “Furthest”= “Far away”, from what? Please consider rephrasing.

L188-194 are printed in a slightly smaller font which makes these lines appear as a part of a figure caption, which I guess they are not.

Discussion:

L252

Consider omitting “meticulous methodology in” and replacing “analysis” by “study design”.

Author Response

Thank you for the review! Please see the attachment.

Corrections made on the second version.  

Best,

Hedayat Rahin

Reviewer 2 Report

The manuscript by Rahin and coworkers describes a crossover study, performed on polyneuropathic patients treated with Trans-spinal Direct Current Stimulation (tSDCS).

The work is well designed and organized.

The Authors should highlight the current limitations of the study, i.e. the low number of patients recruited (n=20), as well as the presence of different aetiological conditions for polyneuropathies. This might be a confounding factor, having an impact on pharmacological therapies that patients need to take, altering the response to tSDCS. Moreover, the same Authors discussed the effects of different BMI, thus subcutaneous fat amounts, on the effectiveness of tSDCS. The current study exploited the same current intensity for all patients, no matter what was their BMI. In the future, this might be taken into account, dividing patients groups by BMI index and adjusting the intensity of stimulations.

Author Response

(The authors gave the same response as above.)

Reviewer 3 Report

The study entitled “Effects of BDNF polymorphism and transcutaneous spinal direct current stimulation in patients with painful polyneuropathy. A randomized, sham-controlled crossover study” aims to evaluate the efficacy of an intervention with TsDCS in polyneuropathy compared to placebo, and to establish the relationship of the possible improvement with a series of predictive factors. This reviewer makes the following suggestions to improve the quality of the manuscript:

TITLE

P1Ln2. The title does not correspond to the objective of the study. BDNF polymorphism is not the only predictor factor that is studied, so it should not be reflected only to it. Reformulate the title according to the purpose of the study.

ABSTRACT

P1Ln11-30. The objective of the study must be reflected in the abstract. Specify the study variables (primary and secondary) and the predictor variables (BDNF, BMI, etc). Write explicitly the type of study and the washout period. Regarding the results, first clarify the effect of tsDCS based on the group (real or sham), and then report the effect based on the predictor variables. It's very confusing. In line 27: Improves pain? In P6Ln205 the authors state that there is no difference in terms of pain.

MATERIALS AND METHODS

P2Ln88. Did the authors follow the CONSORT statement to carry out the study?

P2Ln89-93. Information regarding the number of subjects included in the study or their age, as well as the number of dropouts, must be reported in Results.

P3Ln96. Please list the inclusion and exclusion criteria more clearly under the subheading “Eligibility criteria” (2.2). Was a sample size calculation performed? Otherwise, the power of the study might not be sufficient to give value to the conclusions. Write in limitations.

P3Ln104. Was the study protocol previously registered in any clinical trial database? Reflect on study limitations, with the consequent risk of bias.

P3Ln104. The authors state that they have randomized the subjects to the real group and the sham group. What randomization method was used? Who did it? Was there any method of concealment?

P3Ln107. The study variables (primary and secondary) and the methods used to assess them must be written in another section (“Outcome variables”). Please indicate which investigators were in charge of carrying out the evaluations, interventions, and statistical analysis. Was a medical specialist in charge of recruiting the subjects?

P3Ln108. The abstract mentions that data was collected twice daily. Later in the text it is said that these data are evaluated on days 1, 3, 5 and 7 after the end of the treatment. There must be some consistency throughout the text.

P3Ln110. Why was a one-week washout period decided? Referencing.

P3Ln114. The authors must cite the reference of the document approved by the Ethics Committee, and the hospital/region to which it belongs.

P4Ln123. Is this sham method based on any previous studies? Has it been shown that this short stimulation does not produce any changes at the spinal level? Referencing.

P4Ln133. For a better understanding, subheadings 2.4 and 2.5 should be included in a subheading entitled "Outcome variables" where the study variables (primary and secondary) are detailed, as well as the sociodemographic, clinical variables, etc. that are going to be used as predictors of greater or less successful intervention. Was the evaluator blinded to the group assignment? Pharmacological treatment is not cited by the authors in this section. How were these medicines classified? Please, it is not clear when the data was collected.

P4Ln145. Was any method considered to assess blinding? (James´ or Bang´s Index).

RESULTS

P4Ln160. Why aren't the data presented as mean and SD or percentages for each of the two groups, with their corresponding p?

P4Ln160. List here the number of subjects recruited and finally included, number of dropouts, etc. Include figure 1 here.

P5Ln169. Discuss later why the sham group subjects experienced slight redness in the same way as the real group if they did not receive any type of stimulation.

P5Ln171. Please include the p for each variable in table 2. Idem table 1.

P6Ln185. Please write statistical values ​​for this statement. Same as Ln 191.

P6Ln185. The authors state that both groups were not homogeneous with respect to VAS pain according to BDNF genotype. Can that influence the results? Discuss later.

P6Ln185, 192. The data related to the baseline must be described at the beginning of the Results section.

P6Ln193. Why are subjects with BMI between 25 and 30 not included? What about sleep quality outcome?

P6Ln203. This section 3.2 should precede 3.1, since the latter reports on the predictive factors. There is great confusion in the presentation of the results. It must be clearly stated if real tsDCS was superior to sham tsDCS. Similarly, the authors should consider whether these graphs are the most suitable for these variables.

P6Ln204. Include statistical data, please.

DISCUSSION

P8Ln249. This aspect of “therapy refractive pain” is the first time it appears in the manuscript. Why is it not named in “Eligibility criteria”?

P9Ln259. What MCID value is considered for sleep quality? Same as for pain? Referencing.

P9Ln285. Remove “e.g.”.

P9Ln293. Phenotype better than anatomy. Idem Ln304.

P9Ln296-303. Carefully express these conclusions. Baseline heterogeneity, small sample size, etc. increase the risk of bias in the study.

P10. Include a “Limitations” section.

CONCLUSIONS

P10Ln320. The conclusions do not correspond to the results of the study. Reformulate.

Figure 1. It must belong to the results section, not the method section.

Tables 1 and 2. Legends are missing at the bottom of the table.

Figure 2. Authors should follow a common pattern with figures 3 and 4. In figure 2, negative values ​​go in favor of real current, while in figures 3 and 4 positive values ​​go in favor of real current. It's a mistake? If not, please correct.

Author Response

Thank you for the review! Please see the attachment. 

Best,

Hedayat Rahin

Reviewer 4 Report

The authors carried out a DB CO RCT on the interference of BDNF polymorphism on analgesic response tsDCS for NP.

Methodology seems sound but more infos and improvement in data interpretation are needed.

The authors should report the diagnostic criteria they used for probable NP. The terms NP (ie, Treedes’s criteria) and neuropathy are interchangeable. This is especially important in view of the fact the in 11 out of 20 patients aetiology of pain is unknown (Table1) and that patients number 6 and 18 suffer from rheumatological disease in which a NP diagnosis can be made using NP questionnaires but it is hard when using standard diagnostic criteria for NP.

Unless authors mean  neuropathies of unknown aetiology. In that case they should change Table 1 subheadings. In any event, authors should detail the diagnostic workup to make their findings reproducible. Did all these patients undergo neurophysiological and/or neuroimaging studies?

Besides pain which other neurological signs and symptoms were found in these patients? Which were degrees of disability or quality of life?

Overall the study seems negative as far as tsDCS analgesic properties. Given that patients may perceive difference betwee real stimulation and sham stimulation, it will be difficult to have an actual blind condition in a tsDCS study; the decline of analgesic effect over time can reflect the vanishing of a placebo effect.

In my opinion, the most interesting finding was the effect of tsDCS on sleep: it seems real, persistent and, probably, clinically relevant. BDNF Val66Met polymorphism has been associated to stress, depression and severe depression and response to antidepressants (Youssef et al., 2018; Zhao et al., 2018; Pathak et al., 2022). Did authors score depressive symptoms in these patients? Usually transcranial tDCS is employed for depression.

Line126: …along the spine… can authors have stimulated the vagal nerve at cervical level? Or they stimulated just at lumbar level?

Author Response

(The authors gave the same response as above.)

Round 2

Reviewer 1 Report

I read the revised version of the manuscript and I find that it has improved a lot. All my previous comments have been addressed. I therefore recommend acceptance of the manuscript.

Reviewer 3 Report

The authors have done a magnificent job of reviewing the manuscript.

All suggestions are correctly reviewed. Please consider the editor placing figure 1 in the results section instead of in Methods.

In view of this reviewer, the paper is ready to be published.